# Predicting Immunotherapy Outcomes in Glioblastoma Patients through Machine Learning

**DOI:** 10.3390/cancers16020408

**Published:** 2024-01-18

**Authors:** Guillaume Mestrallet

**Affiliations:** Mount Sinai Hospital, New York, NY 10029, USA; guillaume.mestrallet@mssm.edu

**Keywords:** immune checkpoint, glioblastoma, PD-1, resistance, software

## Abstract

**Simple Summary:**

This scientific study focuses on glioblastoma, a highly aggressive cancer with a poor prognosis. Despite various treatment modalities, including immune checkpoint inhibitors (ICBs), the efficacy of ICBs remains limited, prompting the need for a proactive approach to understand treatment response and resistance. This study involves a thorough analysis of two glioblastoma patient cohorts treated with the Programmed Cell Death Protein 1 (PD-1) blockade. Notably, 60% of the patients exhibited persistent disease progression despite the ICBs. We characterized the immune profiles of these patients with continued cancer progression, revealing multiple defects such as compromised macrophage, monocyte, and T follicular helper responses, impaired antigen presentation, abnormal regulatory T cell (Tregs) responses, and increased expression of immunosuppressive molecules. Using machine learning algorithms, we developed predictive models and software. This computational tool achieved significant success, accurately predicting the progression status of 82.82% of the glioblastoma patients in this study following ICBs based on their unique immune characteristics. In conclusion, this study proposes a personalized approach to immunotherapy in glioblastoma patients. By utilizing patient-specific attributes and computational predictions, we advocate for a paradigm shift towards tailored therapies. This approach has the potential to improve glioblastoma management, offering new possibilities for improved patient care following immunotherapy.

**Abstract:**

Glioblastoma is a highly aggressive cancer associated with a dismal prognosis, with a mere 5% of patients surviving beyond five years post diagnosis. Current therapeutic modalities encompass surgical intervention, radiotherapy, chemotherapy, and immune checkpoint inhibitors (ICBs). However, the efficacy of ICBs remains limited in glioblastoma patients, necessitating a proactive approach to anticipate treatment response and resistance. In this comprehensive study, we conducted a rigorous analysis involving two distinct glioblastoma patient cohorts subjected to PD-1 blockade treatments. Our investigation revealed that a significant portion (60%) of patients exhibit persistent disease progression despite ICB intervention. To elucidate the underpinnings of resistance, we characterized the immune profiles of glioblastoma patients with continued cancer progression following anti-PD1 therapy. These profiles revealed multifaceted defects, encompassing compromised macrophage, monocyte, and T follicular helper responses, impaired antigen presentation, aberrant regulatory T cell (Tregs) responses, and heightened expression of immunosuppressive molecules (TGFB, IL2RA, and CD276). Building upon these resistance profiles, we leveraged cutting-edge machine learning algorithms to develop predictive models and accompanying software. This innovative computational tool achieved remarkable success, accurately forecasting the progression status of 82.82% of the glioblastoma patients in our study following ICBs, based on their unique immune characteristics. In conclusion, our pioneering approach advocates for the personalization of immunotherapy in glioblastoma patients. By harnessing patient-specific attributes and computational predictions, we offer a promising avenue for the enhancement of clinical outcomes in the realm of immunotherapy. This paradigm shift towards tailored therapies underscores the potential to revolutionize the management of glioblastoma, opening new horizons for improved patient care.

## 1. Introduction

Glioblastoma (GBM), the most prevalent primary malignant tumor of the central nervous system, presents a formidable clinical challenge, exhibiting an incidence rate of 3.19 per 100,000 individuals in the United States [1,2]. Despite therapeutic interventions such as surgery, chemotherapy (temozolomide), radiotherapy, and immune checkpoint blockade (ICB), the prognosis remains stark, with a median survival time of merely 15 months, a median progression-free survival of 7 months, and less than 5% of patients surviving beyond 5 years post diagnosis [2,3].

While immune checkpoint blockade, particularly targeting Programmed Cell Death Protein 1 (PD-1), has emerged as a promising avenue, its efficacy in adjuvant settings is limited, with notable exceptions in cases of mismatch repair deficiency [2,4,5,6,7,8,9,10]. Notably, the neoadjuvant administration of pembrolizumab for patients with glioblastoma, followed by continued adjuvant therapy post surgery, has demonstrated a significant extension in the overall survival compared to exclusive adjuvant anti-PD-1 treatment [2]. This approach is associated with the upregulated expression of T cell and IFNγ-related genes and the downregulated expression of cell-cycle-related genes within the tumor, accompanied by fewer monocytes in the bloodstream compared to adjuvant therapy alone. However, even in neoadjuvant settings, resistance to immune checkpoint blockade remains a challenging issue.

Against this backdrop, a comprehensive meta-analysis encompassing glioblastoma patient cohorts assumes a paramount importance in unraveling the intricate mechanisms governing response and resistance to immune checkpoint blockade. The amalgamation of data from diverse cohorts holds promise in delineating optimal targets for the development of combination therapies, thereby enhancing patient outcomes [2,5]. Additionally, the integration of machine learning approaches into the analysis and interpretation of these datasets stands out as a powerful strategy (Figure 1). Building upon successful applications in predicting patient outcomes in lung and gastric cancers [11,12,13], machine learning tools offer the potential to refine the precision of response and resistance prediction to immune checkpoint blockade. This, in turn, holds the key to improving diagnostic accuracy and tailoring therapeutic strategies based on patient-specific characteristics.

To enhance the monitoring of glioblastoma patients undergoing immunotherapy, our proposal involves a two-step approach (Figure 1). First, we aim to identify key immune features associated with either resistance or positive response to therapy. Subsequently, these identified features will serve as input for training machine learning algorithms, enabling the development of personalized prediction models tailored to individual patients based on their unique immune profiles. These machine learning models, once trained, can be applied to predict the likely response of new patients to immunotherapy. In cases where a positive response is anticipated, recommending immunotherapy for that specific patient could prove beneficial. Conversely, if the prediction indicates resistance, it may be more advantageous to consider alternative therapeutic strategies. Additionally, continuous monitoring of the patient’s response evolution would be crucial for optimizing their care. This approach aims to refine treatment decisions, ultimately improving outcomes for glioblastoma patients undergoing immunotherapy.

To better monitor patients with glioblastoma following immunotherapy, we propose to first identify immune features associated with resistance or response to therapy. Then, these features may be used to train machine learning algorithms to predict the personalized responses of patients based on their specific immune features. These algorithms may be used to predict the responses of new patients. If the predicted outcome is a positive response, it may be beneficial to use immunotherapy for this patient. If the predicted outcome is resistance, it may be better to use alternative therapeutic strategies and/or to better monitor the evolution of the response to this patient’s care.

## 2. Material and Methods

### 2.1. RNAseq Datasets and Selection of Cohorts

The patient cohort was selected using the CRI iAtlas Portal [14]. The Cancer Research Institute (CRI) iAtlas is an interactive web platform and includes analytic tools designed for studying interactions between tumors and the immune microenvironment. It enables researchers to explore associations across various genomic characterizations, clinical phenotypes, germline genetics, and responses to immunotherapy. The platform includes modules for immune checkpoint inhibitor analysis, allowing the interactive exploration of biomarkers’ relationship with checkpoint blockade outcomes. The tool harmonizes primary sequencing data from 12 immuno-oncology trials with genomics and clinical data. iAtlas also facilitates the identification of how tumor-intrinsic alterations, such as mutations and copy-number alterations, relate to the immune microenvironment. Initially based on The Cancer Genome Atlas (TCGA) data, iAtlas has expanded to incorporate data from the Pan-Cancer Analysis of Whole Genomes (PCAWG), and efforts are ongoing to include additional immuno-oncology datasets. The platform is a collaborative effort involving the Cancer Research Institute, Sage Bionetworks, the Institute for Systems Biology, and the Vincent Lab at the UNC Lineberger Comprehensive Cancer Center.

We selected the following RNAseq datasets for glioblastoma patients: Zhao 2019—GBM, PD-1 [5], Prins 2019—GBM, and PD-1 [2]. In Zhao’s study, all the patients were treated with the standard therapy of Temozolomide (Merck, Darmstadt, Germany) and radiation before the administration of PD-1 inhibitors [5]. In Prins’s study, Pembrolizumab (Merck) was given (200 mg) by means of intravenous infusion 14 ± 5 d before a scheduled surgical resection [2]. After recovery from surgery, the patients received Pembrolizumab (200 mg) every 3 weeks until either tumor progression or an adverse event requiring study drug discontinuation. Blood samples were obtained every two cycles (6 weeks). RNA was isolated from tumor sections stored in AllProtect tissue reagent (Qiagen, Venlo, The Netherlands) at the time of surgery; peripheral blood mononuclear cells were isolated at the baseline, at the time of surgery, and at cycle 2 of therapy and lysed to obtain RNA and proteins [2]. We used the following group filters: “Progression”, “Drug”, and “GBM”. Non-progressors were defined as patients with mRECIST of partial response, complete response, or stable disease, whereas progressors were those with a progressive disease. Then, we used the ICI Analysis Modules. The current version of the iAtlas Portal was built in R using code hosted at https://github.com/CRI-iAtlas/iatlas-app (accessed on 20 September 2023). Assayed samples were collected prior to immunotherapy. The combined datasets stratified by drug therapy are detailed in Table 1. Among the patients treated with Nivolumab (Bristol Myers Squibb, Seattle, WA, USA), targeting PD1, 11 (40%) exhibited non-progression, while 17 (60%) showed progression. In the case of patients administered Pembrolizumab, another PD1-targeting therapy, 15 (44%) were classified as non-progressors, while 19 (56%) were classified as progressors.

### 2.2. Immune Landscape of Cancer in iAtlas

The inaugural release of iAtlas augmented our analysis with additional insights from The Cancer Genome Atlas (TCGA) Research Network, incorporating a comprehensive dataset encompassing over 10,000 tumor samples and spanning 33 tumor types (referred to here as Immune Landscape) [15]. We compared the immune transcriptional signatures available in the CRI iAtlas portal between progressors and non-progressors following ICB. In more detail, we first compared the immune features between progressors and non-progressors following ICB, which are the transcriptomic signatures corresponding to each immune type (B cells, T cells, macrophages, and others). Then, we compared the immunomodulator features between progressors and non-progressors following ICB, which are the transcriptomic expression corresponding to each immunomodulator available on the CRI iAtlas dataset (PD1, TIM3, LAG3, TIGIT, and others).

### 2.3. Statistics

Statistical significance regarding the observed differences was determined utilizing the independent Wilcoxon *t*-test with a multiple-sample correction. All the presented data are expressed as mean ± SEM, with significance set at a *p*-value below 0.05 (*: *p* < 0.05).

### 2.4. Software Development to Predict Response to PD1 Blockade

Patients from two cohorts underwent anti-PD-1 treatment for glioblastoma. Upon merging the cohorts, progression was assessed based on the specific drug therapy. The study comprised *n* = 28 for Nivolumab and *n* = 34 for Pembrolizumab. Our developed software, coded using Python, HTML, CSS, MySQL, and Django, empowers registered clinicians to either diagnose a new patient or retrieve the diagnosis of a previously registered patient by entering their medical identifier into a form, as detailed in previous studies [16,17]. This innovative tool calculates the probability of patient response to anti-PD1 treatment upon completion of the form, thereby aiding in personalized treatment decisions.

This prediction is based on a RandomForestClassifier, an ensemble-learning algorithm which belongs to the broader category of decision tree classifiers. The primary idea behind a RandomForestClassifier is to create multiple decision trees during the training phase and combine their predictions to achieve a more accurate and robust result. Here is an overview of how a RandomForestClassifier works:Bootstrapped Sampling (Bagging): The algorithm begins by creating multiple subsets of the original dataset through a process called bootstrapped sampling. Each subset is essentially a random sample with a replacement from the original dataset. These subsets serve as the training data for individual decision trees.Decision Tree Construction: A decision tree is built using each of these bootstrapped datasets. At each node of the tree, the algorithm selects a random subset of features to consider for making a split. This randomness helps the algorithm to diversify the trees in the forest.Voting (Classification) or Averaging (Regression): Once all the decision trees are constructed, they “vote” on the class label in the case of classification problems or provide a numerical prediction in the case of regression problems. For classification, the class with the most votes becomes the predicted class. For regression, the average of all the predictions is taken.Ensemble Output: The final output of the RandomForestClassifier is determined by aggregating the individual outputs from all the decision trees. This process helps us to reduce overfitting and increase the model’s generalization ability.

The key characteristics of a RandomForestClassifier are the following:Diversity: The strength of a random forest lies in the diversity of its constituent decision trees. By introducing randomness in the feature selection and dataset creation, each tree becomes unique, contributing different perspectives to the overall prediction.Robustness: Random forests are robust to outliers and noise in the data. The ensemble nature of the model helps it to mitigate the impact of individual decision trees making incorrect predictions.Reduced Overfitting: The combination of multiple trees helps us to overcome overfitting, a common issue with individual decision trees. The ensemble approach tends to yield more stable and reliable predictions.Feature Importance: RandomForestClassifiers can provide information about feature importance, indicating which features are more influential in making predictions. This can be valuable for understanding the underlying dynamics of the dataset.

In summary, a RandomForestClassifier is a powerful and versatile machine learning algorithm that leverages the wisdom of multiple decision trees to make robust and accurate predictions for both classification and regression tasks [18,19,20,21,22,23,24,25,26].

## 3. Results

### 3.1. Progression and Overall Survival of Glioblastoma Patients According to PD1 Blockade in Two Cohorts

We conducted a comprehensive analysis of the progression and overall survival outcomes in glioblastoma patients undergoing immune checkpoint therapy targeting PD1 across two distinct cohorts. The aggregated results, delineated in Table 1, illuminate the correlation between progression status and each checkpoint combination.

For the patients administered Nivolumab, a PD1-targeting therapy, 11 (40%) were classified as non-progressors, while 17 (60%) exhibited progression. Similarly, in the case of patients treated with Pembrolizumab, targeting PD1, 15 (44%) demonstrated non-progression, and 19 (56%) experienced disease progression.

When examining the patients subjected to monotherapy targeting PD1 with disease progression, the overall survival rate remained below 30%, as depicted in Figure 2. Conversely, among the non-progressors following Pembrolizumab, there was a notable improvement, with an overall survival rate reaching approximately 60%. These findings collectively underscore the clinical impact of PD1 blockade therapy on the progression and survival outcomes of glioblastoma patients. Notably, 60% of the patient cohort exhibited robust resistance to PD1 blockade therapy, highlighting the challenges associated with achieving favorable responses in this context.

Patients from two cohorts received anti-PD-1 following glioblastoma. The overall survival was calculated in these cohorts (days): *n* = 28 for Nivolumab and *n* = 34 for Pembrolizumab.

### 3.2. Immune Response and Resistance in Glioblastoma Patients following PD1 Blockade

In our exploration of immune features linked to response and resistance to immune checkpoint therapy in glioblastoma patients, we leveraged the CRI iAtlas to scrutinize the immune landscape in individuals treated with anti-PD1 agents, namely, Nivolumab or Pembrolizumab.

An analysis of the patients’ immune response revealed distinctive profiles in the patients treated with Pembrolizumab. The non-progressors exhibited heightened levels of monocytes and T follicular helpers, while the progressors displayed elevated counts of macrophages, particularly M0, and Tregs (Figure 3). Notably, no significant differences were observed for other immune subsets within the Pembrolizumab cohort and none at all in the Nivolumab cohort (Figure 3 and Appendix A). This may be explained by cohort size and the fact that the drug was used in adjuvant settings in Zhao’s study and not in neoadjuvant settings as had been the case in Prins’ study.

Exploring the expression of immunomodulatory molecules, we found that the progressors following Pembrolizumab administration demonstrated increased expression of MHC molecules (HLA-DRB1, HLA-DQA1, HLA-DRB5, HLA-DQB1) compared to the non-progressors. This point towards a potential role of defects in antigen presentation in promoting anti-tumor immune resistance (Figure 4). Intriguingly, the upregulation of ITG2B, a gene linked to T cell adhesion, was observed in the Pembrolizumab progressors, indicating that T cell responses alone might not be sufficient for a favorable outcome. Importantly, the progressors following Pembrolizumab treatment exhibited heightened expression of immunosuppressive molecules such as TGFB, IL2RA, and CD276. In contrast, the progressors following Nivolumab administration displayed lower expression of BTN3A1, a gene involved in T cell activation, IL4, and ARG1. No differences were observed in the expression of other immunoregulatory molecules and immune checkpoints across both cohorts (including PD1, TIGIT, TIM3, LAG3, EDNRB, TLR4, VSIR, CD40, TNFRSF, CD28, ICOS, VTCN1, CD70, CX3CL1, ENTPD1, GXMA, HMGB1, ICOSLG, VEGF, KIR, IFN genes, interleukins, MICA, and other HLA genes) (Appendix A). The disparities observed in one cohort did not mirror those in the other cohort (Figure 3, Figure 4, Appendix A).

Overall, the glioblastoma patients experiencing cancer progression following anti-PD1 therapy displayed distinctive immune signatures, marked by deficiencies in macrophage, monocyte, and T follicular helper responses, impaired antigen presentation, skewed Tregs response, and heightened expression of immunosuppressive molecules (TGFB, IL2RA, and CD276). These insights contribute to a deeper understanding of the intricate interplay between immune elements and therapeutic outcomes in the context of PD1 blockade.

A total of 62 patients received anti-PD-1 following glioblastoma: *n* = 28 for Nivolumab (Zhao) and *n* = 34 for Pembrolizumab (Prins). The patients’ immune response was measured using CRI iAtlas, with * = *p* < 0.05, according to Wilcoxon’s *t*-test.

The patients’ immunomodulatory molecule expression was measured using CRI iAtlas, with * = *p* < 0.05, according to Wilcoxon’s *t*-test.

### 3.3. Software Prediction of Glioblastoma Patients’ Response to PD1 Blockade

In our pursuit of personalized response prediction for glioblastoma patients undergoing PD1 blockade, we employed a RandomForestClassifier trained on a dataset enriched with features identified as differentially expressed between progressors and non-progressors post PD1 blockade (Figure 3 and Figure 4). The model demonstrated an overall accuracy of approximately 81.82%, signifying its ability to correctly predict the progression status for the majority of data points in the test set (Figure 5).

Delving into more granular performance metrics, the classification report offers insights into precision, recall, and F1-score for each class, namely, “Non_Progressor” and “Progressor”. Precision, denoting the accuracy of positive predictions, achieved a score of 100% for the “Non_Progressor” class, indicating that all the positive predictions for this class were correct. For the “Progressor” class, precision stood at 80%, implying that 80% of the positive predictions for this class were accurate. Moving to recall, which gauges the model’s ability to identify all the relevant instances of a class, “Non_Progressor” scored 33%, indicating that only 33% of the actual “Non_Progressor” instances were accurately identified. In contrast, “Progressor” achieved a recall of 100%, signifying that all the actual “Progressor” instances were correctly identified. The F1-score, representing the harmonic mean of precision and recall, strikes a balance between the two metrics. “Non_Progressor” yielded an F1-score of 0.50, while “Progressor” achieved a high F1-score of 0.89, underscoring the model’s robust performance in identifying instances of disease progression. Support, indicating the number of samples in each class within the test set, revealed that “Non_Progressor” comprised three samples, while “Progressor” encompassed eight samples.

In summary, our RandomForestClassifier-based model excelled in predicting the progression status of glioblastoma patients, achieving a success rate of 82.82%. These findings highlight the potential of our software to contribute valuable insights regarding the prognosis of patients undergoing PD1 blockade.

Utilizing the methodologies described in this study, we applied predictive analyses to anticipate the response of a hypothetical patient to PD1 blockade, as illustrated in Figure 6. Leveraging her genetic characteristics, our algorithm projected an 82.82% probability that the patient would be classified as a “Progressor” following anti-PD1 therapy. Consequently, considering this high likelihood of progression, alternative therapeutic options may warrant careful consideration for the optimal management of this particular patient.

Figure 6 displays the glioblastoma patient data collection form as well as the global probability of the patient responding to different immune checkpoint blockades and the personalized probability of the patient to respond to the immune checkpoint blockade based on her genetic characteristics and machine learning prediction.

## 4. Discussion

In our analysis of two cohorts tracking the progression of glioblastoma patients undergoing immune checkpoint blockade (ICB), 60% of them exhibited robust resistance to PD1 blockade therapy. Those experiencing cancer progression following anti-PD1 therapy demonstrated distinct immunological profiles, marked by deficiencies in macrophage, monocyte, and T follicular helper responses, compromised antigen presentation, aberrant Tregs response, and elevated expression of immunosuppressive molecules such as TGFB, IL2RA, and CD276. Notably, CD276, known to regulate cell proliferation, invasion, and migration in cancers, emerged as a key factor in this resistance landscape [27]. It is expressed on various immune cells and in a GBM cell subpopulation able to leave the tumor and invade the subventricular zone [28]. High IL2RA expression, observed in the alpha chain of the interleukin 2 receptor complex on mature T cells, has been shown to correlate with unfavorable survival outcomes in pancreatic ductal adenocarcinoma patients [29]. TGFB, a pivotal mediator in various biological processes, has also been implicated in resistance to immunotherapy [30]. In our study, progressors following Nivolumab treatment exhibited a lower expression of BTN3A1, a molecule coordinating αβ and γδ T cells [31]. Ongoing clinical trials hold promise for providing more precise insights into immune resistance patterns following ICB in these patients [4].

To enhance our ability to predict ICB responses in glioblastoma patients, we developed machine learning approaches. Utilizing the CRI iAtlas dataset, a RandomForestClassifier trained on features identified as being differentially expressed between progressors and non-progressors post anti-PD-1 blockade achieved a notable success rate, accurately predicting the progression status in 82.82% of the patients. While our model demonstrated good precision and recall for the “Progressor” class, indicating effective identification of these cases, there is room for improvement in identifying “Non_Progressors”, as suggested by the lower recall for that class. Expanding the size of the training dataset stands out as a potential strategy to refine the model’s predictive capabilities and may reduce the risk of overfitting.

Using bioinformatics to identify biomarkers is not new [32,33], but the advantages of our machine learning approach are as follows: it is more specific to glioblastoma patients, and it can combine a large number of features to improve its prediction compared to other algorithms. Moreover, our study emphasizes the use of simple transcriptomics features instead of more complex datasets like single-cell RNA sequencing (scRNAseq). Thus, it may be a cost-effective alternative, and it may make predictive tools more accessible to hospitals with limited financial resources.

In conclusion, we introduced a software application that incorporates our machine learning approach to predict patient responses to the immune checkpoint blockade. This software computes the probability of being a progressor or a non-progressor to the PD-1 blockade based on a patient’s specific immune characteristics. The development of such machine learning tools tailored to individual patient characteristics holds great promise for facilitating more personalized and relevant treatment strategies.

## 5. Conclusions

Our approach advocates for the personalization of immunotherapy in glioblastoma patients. By harnessing patient-specific immune and genetic attributes and computational predictions, we offer a promising avenue for the enhancement of clinical outcomes following immunotherapy. This paradigm shift towards tailored therapies underscores the potential to revolutionize the management of glioblastoma, opening new horizons for improved patient care.

## Figures and Tables

**Figure 1 cancers-16-00408-f001:**
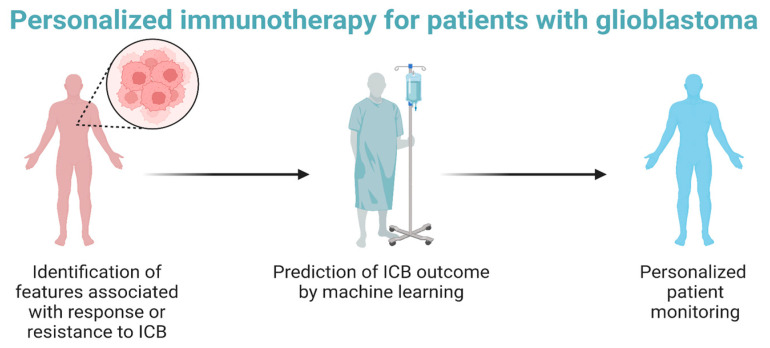
Schematic diagram of the proposed approach to monitor patients with glioblastoma.

**Figure 2 cancers-16-00408-f002:**
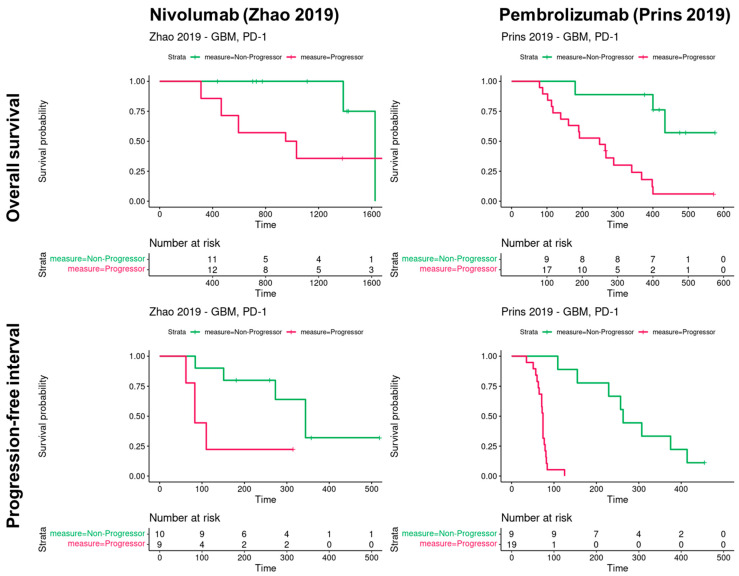
Overall survival of glioblastoma patients according to PD1 blockade in two cohorts [2,5].

**Figure 3 cancers-16-00408-f003:**
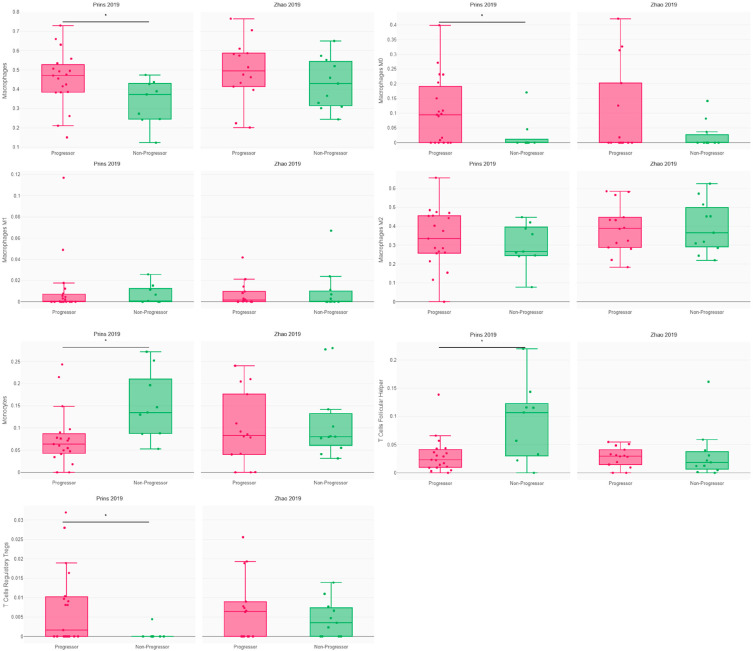
Immune response in glioblastoma patients according to progression following PD1 blockade [2,5]. * = *p* < 0.05.

**Figure 4 cancers-16-00408-f004:**
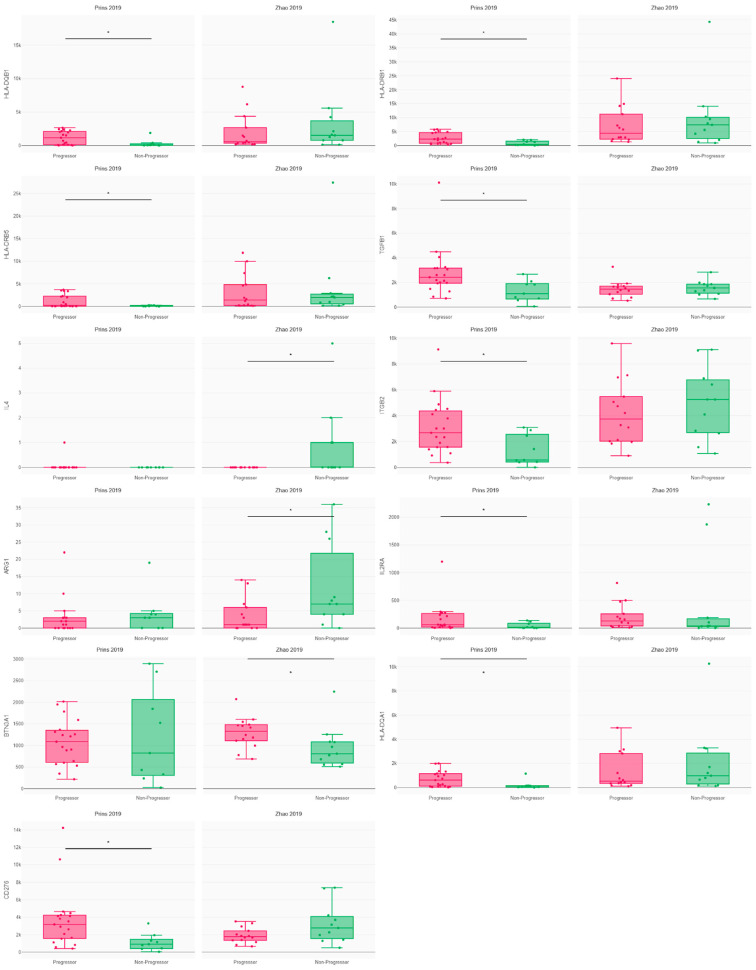
Immunomodulatory molecule expression in glioblastoma patients according to progression following PD1 blockade [2,5]. * = *p* < 0.05.

**Figure 5 cancers-16-00408-f005:**
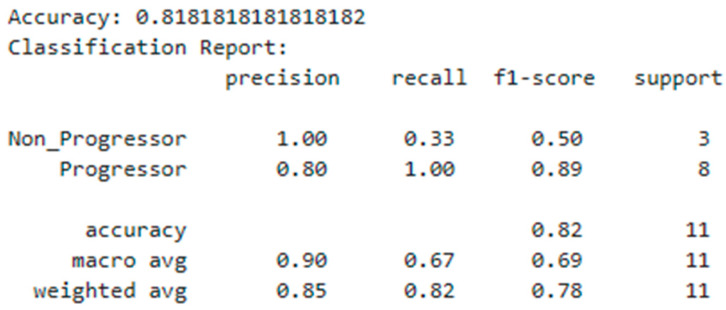
Performance of the algorithm predicting glioblastoma patients’ responses to PD1 blockade.

**Figure 6 cancers-16-00408-f006:**
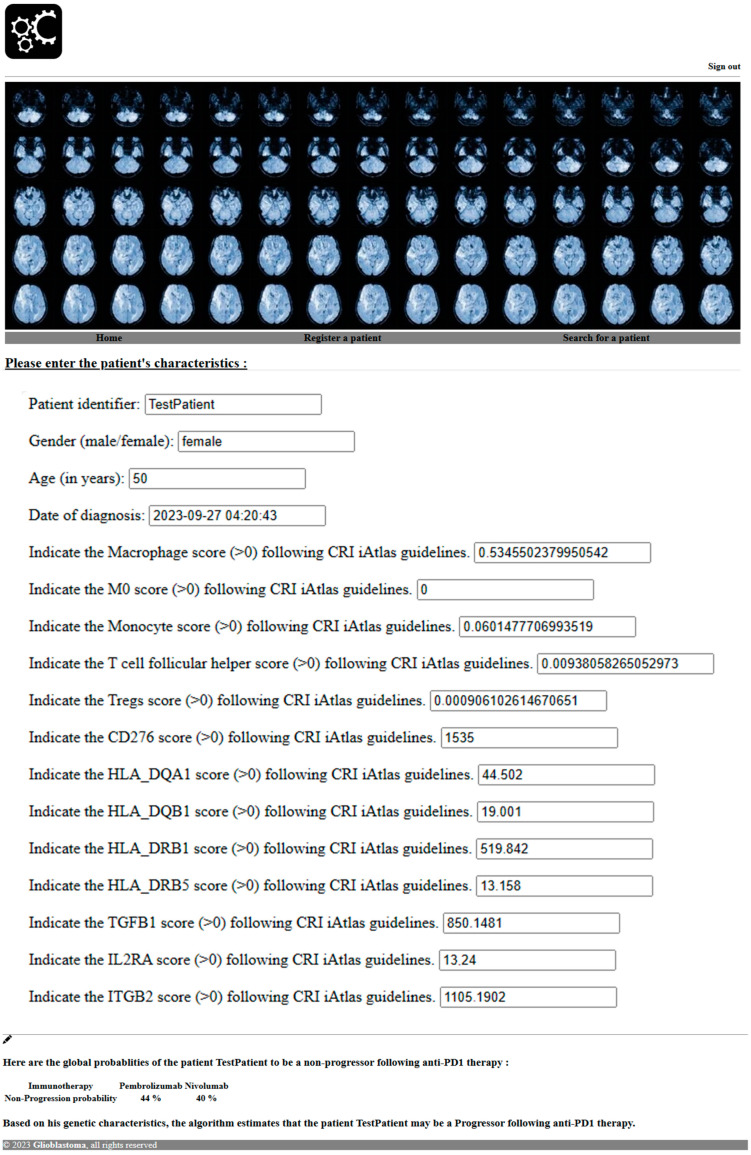
Software prediction of glioblastoma patient response to PD1 blockade.

**Table 1 cancers-16-00408-t001:** Response of patients with glioblastoma to PD1 blockade.

Drug	Nivolumab	Pembrolizumab
Target	PD1	PD1
Non-progressors (nb)	11	15
Progressors (nb)	17	19
Non-progressors (%)	40	44
Progressors (%)	60	56

## Data Availability

Data are available on CRIiAtlas website https://isb-cgc.shinyapps.io/iatlas/ and code is available on Github https://github.com/gmestrallet/Cancers_2024_16.

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
