# Peer review of "Predicting Immunotherapy Outcomes in Glioblastoma Patients through Machine Learning"

_cancers, 2024, doi:10.3390/cancers16020408_

Round 1

Reviewer 1 Report

Comments and Suggestions for Authors

When analysing clinical data it is important to represent them more thoroughly.

Author Response

Thanks to reviewer 1 for his/her suggestions. 

‘When analysing clinical data it is important to represent them more thoroughly.’

The representation of the clinical data was improved as suggested and we added the progression free interval.

Best regards

Guillaume

Reviewer 2 Report

Comments and Suggestions for Authors

This study delves into the challenging landscape of glioblastoma, a highly aggressive cancer with a grim prognosis and limited success in existing therapeutic approaches. Focusing on the efficacy of immune checkpoint inhibitors (ICB), specifically PD-1 blockade treatments, this work involves a meticulous analysis of two distinct glioblastoma patient cohorts. They found that 60% of patients displayed persistent disease progression despite undergoing ICB intervention. To unravel the mechanisms of resistance, the immune profiles of patients experiencing continued cancer progression post-anti-PD1 therapy are comprehensively characterized. The intricate examination reveals a spectrum of defects, ranging from compromised macrophage, monocyte, and T follicular helper responses to impaired antigen presentation, aberrant Treg responses, and heightened expression of immunosuppressive molecules (TGFB, IL2RA, and CD276).

(1) There are a lot of similar bioinformatics work revealing biomarkers. I would recommend the authors discuss their method advantages over the other bioinformatics work (e.g. PMID: 35173534) that also uses survival analysis to help reveal potential biomarkers.

(2) The analysis methods are lack of details. The authors need to provide more descriptions on the methods.

(3) I would recommend the authors to polish figures with high resolution and consistent font size.

(4) I recommend the authors to include some discussions on related studies using different omics data (PMID: 33461059; PMID: 35284940), which helps expand the scope of the study.

Comments on the Quality of English Language

N/A

Author Response

Thanks to reviewer 2 for his/her suggestions. 

‘This study delves into the challenging landscape of glioblastoma, a highly aggressive cancer with a grim prognosis and limited success in existing therapeutic approaches. Focusing on the efficacy of immune checkpoint inhibitors (ICB), specifically PD-1 blockade treatments, this work involves a meticulous analysis of two distinct glioblastoma patient cohorts. They found that 60% of patients displayed persistent disease progression despite undergoing ICB intervention. To unravel the mechanisms of resistance, the immune profiles of patients experiencing continued cancer progression post-anti-PD1 therapy are comprehensively characterized. The intricate examination reveals a spectrum of defects, ranging from compromised macrophage, monocyte, and T follicular helper responses to impaired antigen presentation, aberrant Treg responses, and heightened expression of immunosuppressive molecules (TGFB, IL2RA, and CD276).

(1) There are a lot of similar bioinformatics work revealing biomarkers. I would recommend the authors discuss their method advantages over the other bioinformatics work (e.g. PMID: 35173534) that also uses survival analysis to help reveal potential biomarkers.’

We discussed our method advantages compared to this work as suggested.

‘(2) The analysis methods are lack of details. The authors need to provide more descriptions on the methods.’

More details and descriptions were added on the methods section as suggested.

‘(3) I would recommend the authors to polish figures with high resolution and consistent font size.’

Figures and font size were improved as suggested.

‘(4) I recommend the authors to include some discussions on related studies using different omics data (PMID: 33461059; PMID: 35284940), which helps expand the scope of the study.’

Discussion on these studies was added as suggested.

Best regards

Guillaume

Reviewer 3 Report

Comments and Suggestions for Authors

Dear Authors,

Thank you so much for reviewing this manuscript “Title: Enhancing immunotherapy outcomes in glioblastoma through predictive machine learning”. I give some of the following suggestions to improve the readability and understanding of your studies.

1.     Please add novelty to your works with schematic diagrams in the introduction sections.

2.     Please add a conclusion section and explain the overall conclusion.

3.     Please check the grammatical mistakes and follow the designated journals' proper references and reference style.

4.     Please add more figures for every section of your manuscript.

Thank you so much.

Best Regards

Author Response

Thanks to reviewer 3 for his/her suggestions. 

‘1.     Please add novelty to your works with schematic diagrams in the introduction sections.’

Schematic diagrams showing novelty were added in the introduction.

‘2.     Please add a conclusion section and explain the overall conclusion.’

Conclusion was added in the new version of the manuscript.

‘3.     Please check the grammatical mistakes and follow the designated journals' proper references and reference style.’

References were modified as suggested.

‘4.     Please add more figures for every section of your manuscript.’

Figures were added in the manuscript.

Best regards

Guillaume

Round 2

Reviewer 1 Report

Comments and Suggestions for Authors

In new revision the author did not answer all my comments and questions. 

The title needs to be corrected and more accurate because this title completely does not correspond to the results. Machine learning is currently not helping patients with GBM at all. The «individual approach» is not visible. Moreover, the results of treatment when blocking the PD-1 receptor are still almost hopeless being seen from the manuscript.

New Figure 1 is not usefull.

The article does not contain information about the PD-1 acronym and its role in the immuno- and pathogenesis of GBM.

What is the difference between Nivolumab and Pembrolizumab? Why were taken the results of using these particular drugs?

 In Prins study (Figure 2), out of 7 compared parameters, 5 have statistically significant differences, and in Zhao study there are no differences. How can this data be explained?

In Figure 4, the results of the two studies do not match at all, although «progressors» and «non-progressors» do not differ - about 60%.

 May be it can be explained by different doses of anti-PD-1 drugs, also glucocorticoids and temozolomide, as well as the timing of sampling for studying immune markers, were likely important. Drug administration can influence and modulate the expression of a number of molecules on the surface of T-lymphocytes and macrophages. It is therefore important to indicate when the samples are taken, whether before or during treatment. And then it is possible to compare the data.

 Why the CB276 molecule turned out to be very significant for characterizing the immune status of patients with GBM. 

Author Response

Thanks to reviewer 1 for his/her new suggestions.

‘In new revision the author did not answer all my comments and questions.’

Please apologize if we missed some of the previous comments. On our interface, the only comment that appeared was the one that we addressed. We revised the manuscript following all the comments below.

‘The title needs to be corrected and more accurate because this title completely does not correspond to the results. Machine learning is currently not helping patients with GBM at all. The «individual approach» is not visible. Moreover, the results of treatment when blocking the PD-1 receptor are still almost hopeless being seen from the manuscript.’

The title of the manuscript was modified as suggested. In this manuscript, we showed that identifying immune features associated with response or resistance to PD1 blockade allow us to predict patient response by machine learning with an accuracy of 82%. In cases where a positive response is anticipated, recommending immunotherapy for that specific patient could prove beneficial. Conversely, if the prediction indicates resistance, it may be more advantageous to consider alternative therapeutic strategies. Additionally, continuous monitoring of the patient's response evolution would be crucial for optimizing their care.

‘New Figure 1 is not usefull.’

This figure was not in the first draft but was requested by reviewer 3 in the review process. Please let us know if we should keep it or remove it.

‘The article does not contain information about the PD-1 acronym and its role in the immuno- and pathogenesis of GBM.’

The acronym "PD-1" stands for "Programmed Cell Death Protein 1." PD-1 is a cell surface receptor that plays a crucial role in regulating the immune system. It is primarily found on the surface of certain immune cells, including T cells. Neoadjuvant administration of pembrolizumab for patients with glioblastoma, followed by continued adjuvant therapy post-surgery, has demonstrated a significant extension in overall survival compared to exclusive adjuvant anti-PD-1 treatment (doi: 10.1038/s41591-018-0337-7). It was added in the new version of the manuscript.

‘What is the difference between Nivolumab and Pembrolizumab? Why were taken the results of using these particular drugs?’

Nivolumab and Pembrolizumab are both immune checkpoint inhibitors, specifically targeting the programmed cell death protein 1 (PD-1) receptor. There are structural differences between the two antibodies in terms of how they bind to PD-1. The Pembrolizumab epitope region shows a much greater overlap with the PD-L1 binding site than the epitope region of Nivolumab. As the only datasets available to perform the analysis with immune features and machine learning involve these drugs, we chose these two datasets.

‘ In Prins study (Figure 2), out of 7 compared parameters, 5 have statistically significant differences, and in Zhao study there are no differences. How can this data be explained?’

It may be explained by the fact that there are less patients in the Zhao study, and because the drug was used in adjuvant settings and not in neoadjuvant settings as in Prins study. This was clarified in the manuscript.

‘In Figure 4, the results of the two studies do not match at all, although «progressors» and «non-progressors» do not differ - about 60%.

May be it can be explained by different doses of anti-PD-1 drugs, also glucocorticoids and temozolomide, as well as the timing of sampling for studying immune markers, were likely important. Drug administration can influence and modulate the expression of a number of molecules on the surface of T-lymphocytes and macrophages. It is therefore important to indicate when the samples are taken, whether before or during treatment. And then it is possible to compare the data.’

We agree with the fact that different doses of anti-PD-1 drugs, also glucocorticoids and temozolomide, as well as the timing of sampling for studying immune markers may explain the differences. It may also be explained by the fact that the drug was used in adjuvant settings in the Zhao study and not in neoadjuvant settings as in Prins study. In the Zhao study, all patients were treated with the standard therapy of temozolomide and radiation before the administration of PD-1 inhibitors. In the Prins study, Pembrolizumab was given (200 mg) by intravenous infusion 14 ± 5 d before scheduled surgical resection. After recovery from surgery, patients received Pembrolizumab (200 mg)  every 3 weeks until either tumor progression or an adverse event requiring study drug discontinuation. Blood samples were obtained every two cycles (6 weeks). RNA was isolated from tumor sections stored in AllProtect tissue reagent (Qiagen) at the time of surgery; peripheral blood mononuclear cells were isolated at baseline, time of surgery and at cycle 2 of therapy and lysed to obtain RNA and protein. This was clarified in the manuscript.

 ‘Why the CB276 molecule turned out to be very significant for characterizing the immune status of patients with GBM.’

CD276 is known to regulate cell proliferation, invasion, and migration in cancers, and emerged as a key resistance factor (doi: 10.3389/fonc.2021.654684). It is expressed on various immune cells and in a GBM cell subpopulation able to leave the tumor and invade the subventricular zone (https://doi.org/10.1186/s40478-021-01167-w). This was clarified in the manuscript.

Best regards

Guillaume

Reviewer 2 Report

Comments and Suggestions for Authors

I have no more concerns on this manuscript. It is ready to be accepted. 

Author Response

Thank you.

Round 3

Reviewer 1 Report

Comments and Suggestions for Authors

The title of the article is still not entirely specific.